# The Role of Thymine DNA Glycosylase in Transcription, Active DNA Demethylation, and Cancer

**DOI:** 10.3390/cancers14030765

**Published:** 2022-02-01

**Authors:** Oladapo Onabote, Haider M. Hassan, Majdina Isovic, Joseph Torchia

**Affiliations:** 1Department of Biochemistry, Western University, London, ON N6A 5C1, Canada; oonabot@uwo.ca (O.O.); hhassa3@uwo.ca (H.M.H.); 2Department of Oncology, London Regional Cancer Program and the Lawson Health Research Institute, London, ON N6A 5W9, Canada; mbambego@uwo.ca

**Keywords:** Thymine DNA Glycosylase, transcription, coactivator, active DNA demethylation, chromatin reorganization, tumor suppressor, cancer

## Abstract

**Simple Summary:**

Thymine DNA Glycosylase (TDG) is a DNA repair protein that plays an important role in gene regulation. Recent studies have shown that TDG interacts with various transcription factors to activate target genes. TDG also functions in a pathway known as active DNA demethylation, which removes 5-mC from DNA and replaces it with unmethylated cytosine. In this review, we summarize the various functions of TDG in gene regulation as well as the physiological relevance of TDG in cancer.

**Abstract:**

DNA methylation is an essential covalent modification that is required for growth and development. Once considered to be a relatively stable epigenetic mark, many studies have established that DNA methylation is dynamic. The 5-methylcytosine (5-mC) mark can be removed through active DNA demethylation in which 5-mC is converted to an unmodified cytosine through an oxidative pathway coupled to base excision repair (BER). The BER enzyme Thymine DNA Glycosylase (TDG) plays a key role in active DNA demethylation by excising intermediates of 5-mC generated by this process. TDG acts as a key player in transcriptional regulation through its interactions with various nuclear receptors and transcription factors, in addition to its involvement in classical BER and active DNA demethylation, which serve to protect the stability of the genome and epigenome, respectively. Recent animal studies have identified a connection between the loss of *Tdg* and the onset of tumorigenesis. In this review, we summarize the recent findings on TDG’s function as a transcriptional regulator as well as the physiological relevance of TDG and active DNA demethylation in cancer.

## 1. Introduction

The methylation of cytosine at the 5th carbon (5-mC) is a prevalent form of DNA modification in mammals that is essential for various biological processes, such as X-chromosome inactivation, genomic imprinting, cell differentiation, and the suppression of mobile genetic elements [1]. In addition, aberrant DNA methylation is a common molecular lesion in cancer, usually causing global hypomethylation and locus-specific hypermethylation of tumor suppressor genes. 5-mC occurs predominantly in the context of CpG islands, which are short, interspersed, CG-rich regions of DNA that were originally identified as an unmethylated fraction of mouse genomic DNA [2]. CpG islands are typically 500–3000 base pairs in length and are found within or proximal to nearly half of the promoters of mammalian genes [3]. For example, all housekeeping genes are associated with promoter CpG islands that are enriched in permissive histone marks, contain multiple transcription factor-binding sites, and are typically unmethylated [4]. CpG islands can also be found within enhancer, intergenic, and intronic regions and are often associated with transcriptional regulation of associated genes.

5-mC marks are established during embryonic development by the de novo DNA methyltransferases 3a/3b (DNMT3a/3b), which catalyze the transfer of a methyl group from S-adenosylmethionine (SAM) to the C5 carbon of cytosine [5]. The 5-mC mark is faithfully reproduced over successive replication cycles by the maintenance DNA methyltransferase 1 (DNMT1) in association with the co-factor Ubiquitin-like containing PHD and RING finger domains 1 (UHRF1) [6]. Although 5-mC has long been considered a relatively stable epigenetic mark, several studies have established that DNA methylation, like histone modifications, is dynamic [7,8,9]. In mammals, there are two basic mechanisms involved in the removal of the 5-mC mark: passive and active DNA demethylation. In passive DNA demethylation, the methylated DNA is diluted over successive replication cycles by the deactivation or nuclear exclusion of maintenance DNMT1 or its associated co-factor UHRF1. This is evident during embryonic development, where the maternal genome undergoes passive DNA demethylation by the nuclear exclusion of oocyte specific UHRF1 [10]. Alternatively, 5-mC can be removed in a process called active DNA demethylation that is independent of the cell cycle. Rather, this process is dependent on the oxidative function of the ten eleven translocation (TET) family of proteins, of which there are three functional paralogs (TET1, TET2, and TET3) that exist in mammals due to a gene triplication event that occurred in jawed vertebrates [11,12]. In this process, TETs recognize and successively oxidize 5-mC to 5-hydroxymethylcytosine (5-hmC), and then to 5-formylcytosine (5-fC) and 5-carboxylcytosine (5-caC). 5-fC and 5-caC are then specifically recognized and excised by the base excision repair (BER) protein Thymine DNA Glycosylase (TDG). This generates an apurinic/apyrimidinic (AP) site [13], which is repaired by the combined actions of AP endonuclease 1, DNA Polymerase β and XRCC1-DNA Ligase IIIα complex [14]. This cycle of methylation and active DNA demethylation involving de novo DNMTs, TETs, and BER proteins affords a powerful route to transcriptional manipulation in a replication-independent manner (Figure 1).

In this review, we will first discuss the role of TDG in transcriptional regulation and in active DNA demethylation. We also discuss the recent work using animal models, which establishes a connection between TDG and tumorigenesis in vivo. A large body of evidence has corroborated the enzymatic function of TDG as well as its scaffolding activities in gene regulation. In addition, the physiological relevance of TDG and active DNA demethylation are becoming clearer since a disruption in these processes leads to symptoms of insulin resistance, loss of bile acid homeostasis, and ultimately cancer. The goal of this review is to provide the reader with an overview of the mechanistic work that has been documented on the role of TDG in gene regulation as a framework for understanding its physiological relevance in cancer (a list of all the abbreviations used in this review can be found at the end of the main text).

## 2. The Role of TDG in Transcriptional Regulation

The human *Tdg* gene consists of 10 exons spanning a 23 Kbp region and encodes for a 410 amino acid protein consisting of a centrally positioned catalytic domain flanked by an amino and carboxy terminus, which are lysine-rich and confers important regulatory functions. TDG belongs to the mammalian uracil DNA glycosylase (UDG) superfamily, which all share a common α/β fold [15]. TDG was originally discovered in HeLa cell extracts as a BER enzyme that catalyzes the excision of U:G and T:G mismatches [16]. TDG was then found to glycosylate a variety of mismatched pyrimidine bases, as well as oxidized/halogenated bases such as thymine glycol (Tg), 5-formyluracil (fU), 5-fluorouracil (5-FU), 5-chlorouracil (ClU), 5-hydroxyuracil (5-OHU), 5-hydroxymethyluracil (5-hmU), 3,N^4^-ethenocytosine (εC), 5-hydroxycytosine (5-OHC), 7,8-dihydro-8-oxoadenine (8oxoA), and 5-bromocytosine (BrC) [17,18,19,20,21,22,23,24]. Recent developments have tied TDG to transcriptional regulation by participating in co-activator complex assembly and active DNA demethylation [25]. Much of the evidence supporting a role for TDG in transcriptional regulation is derived from studies demonstrating its interaction with nuclear hormone receptors. Early reports demonstrated a direct interaction between TDG and the retinoic acid receptor α (RARα) and retinoid X receptor α (RXRα) [26,27]. TDG interacts with RARα/RXRα via its catalytic domain in a ligand-dependent manner, thereby enhancing the affinity of RARα for its response element at target genes [27,28]. Ligand binding was also found to trigger the recruitment of additional co-activators, such as CREB-binding protein (CBP) or its related family member p300, that form a ternary complex with TDG and the RARα [29,30]. Importantly, loss of *Tdg* was found to be embryonic lethal at E11.5 that resulted partly from a dysregulation in retinoic acid signaling [31,32]. Furthermore, the differentiation of mouse embryonic stem cells (mESCs) in response to retinoic acid was inhibited in *Tdg*-null mESCs [31]. Mechanistically, TDG was shown to maintain DNA methylation homeostasis and facilitate the recruitment of CBP/p300 and other co-activators at retinoid-dependent target genes. Surprisingly, *Tdg* knockout also prevented the reprogramming of mouse embryonic fibroblasts to induced pluripotent stem cells, demonstrating the requirement of TDG and active DNA demethylation in cell lineage conversion [33].

TDG has also been shown to interact with other members of the nuclear receptor superfamily, including the androgen receptor (AR), glucocorticoid receptor (GR), progesterone receptor (PR), vitamin D3 receptor (VDR), peroxisome proliferator-activated receptor (PPAR), thyroid hormone receptor (TR), and the estrogen receptor (ER) [27,28,34,35,36]. Utilizing a combination of functional and genomic analysis, the involvement of TDG in ER signaling has recently been documented. ER is a ligand-dependent nuclear receptor overexpressed in many breast cancers and is the target of endocrine-based cancer therapies. TDG was shown to localize at a subset of enhancers of ER target genes in an estrogen-dependent manner. Importantly, approximately half of the TDG binding sites characterized were found to overlap with E2-mediated ER binding [37]. Many of the enhancers occupied by TDG and ER were found to actively transcribe enhancer RNAs (eRNAs) and facilitate a 3-dimensional chromatin reorganization to bring promoter and enhancer element in proximity at target genes. Surprisingly, TDG-dependent eRNA transcription and chromatin reorganization were found to be essential for gene expression of some ER target genes.

To corroborate the effects of TDG on chromatin reorganization, a recent study has shown that TDG has the capacity to alter chromatin structure directly through its physical interactions with DNA [38]. Nucleosome array experiments demonstrated that TDG can decondense or open individual chromatin fibers through its interactions with linker DNA. Remarkably, TDG also promotes condensation through long-range interactions between fibers resulting in oligomerization into higher-order chromatin structures. The terminal domains of TDG are critical for this function and appear to have opposing roles during chromatin condensation. TDG mediates chromatin condensation through its amino terminal domain, whereas its carboxyl terminal domain has an antagonizing effect in the process. Moreover, the authors showed that TDG-mediated chromatin condensation can be reversed by growth arrest and DNA damage-inducible protein alpha (GADD45a), providing evidence that TDG’s interactions with GADD45a and other interacting proteins influences its ability to dynamically control chromatin architecture. Altogether, this comprehensive interaction network depicts TDG as a potential scaffold protein and an important player in protein complex stability at target genes.

## 3. TDG in Active DNA Demethylation

Numerous studies have shown that TDG efficiently excises 5-fC and 5-caC oxidation products of 5-mC generated by TET enzymes, and it is TDG’s role in active DNA demethylation, which most likely accounts for the embryonic lethality of *Tdg* knockout mice [39,40,41,42]. Biochemical reconstitution studies using purified recombinant proteins have demonstrated a direct interaction between TET1 and TDG [43]. Furthermore, the TET/TDG complex was highly active and capable of initiating active DNA demethylation in vitro. In the presence of additional BER factors, active DNA demethylation is then completed to correctly re-establish unmodified cytosine on both strands in a sequential manner. TET1 and TDG have also been found to interact physically and are targeted to chromatin by GADD45a [43,44]. GADD45a is a multi-faceted nuclear protein, which has been implicated in DNA demethylation, DNA repair, and genomic stability. Overexpression of GADD45a leads to global DNA demethylation in the presence of TDG and TET proteins, demonstrating that GADD45a enhances DNA demethylation by TDG [45].

The oxidized 5-mC derivatives generated by TETs act as intermediaries for active DNA demethylation and can also accumulate at specific regions throughout the genome. In normal tissues/cell-types, 5-hmC is more abundant than 5-fC/5-caC (>10-fold), as TETs convert 10% of 5-mC to 5-hmC, and only a subset (1–10%) of 5-hmC is converted to 5-fC/5-caC [7]. Although 5mC, 5hmc, and 5fC are all substrates for TET mediated oxidation, they appear to exhibit different substrate and/or catalytic activities. Enzyme kinetic studies suggest that conversion of 5mC to 5hmC is faster than 5hmC to 5fC and 5fC to 5caC. This implies that TET/TDG-mediated oxidation may stall at the 5-hmC step [7].

Genome-wide mapping experiments have shown that 5-hmC is enriched at: (1) promoters that have low CpG density and/or associated with bivalent domains, which are regions that contain both activating and repressive histone marks, typically found in developmental genes that are repressed in ESCs but activated during differentiation, (2) gene bodies of actively transcribed genes, and (3) distal regulatory elements including enhancers, insulators, and regions flanking transcription factor binding sites [46]. The 5-fC and 5-caC metabolites also become enriched within active enhancers, exons, as well as active promoters containing the H3K4me3 chromatin mark, suggestive of an active DNA demethylation process associated with actively transcribing genes [47,48]. Notably, in ESCs, 5-fC/5-caC enrichment increases with the level of promoter accessibility and coincides with the genomic localization of TDG and TET proteins [47].

At a molecular level, 5-fC/5-caC have been shown to impart changes to the physical properties of DNA such as increased DNA flexibility. This can affect supercoiling and packaging of DNA, which can influence gene expression by establishing distinct regulatory regions that directly control the recruitment of specific proteins [49,50]. Interacting proteins specific for each oxidized derivative have been identified that function in DNA repair, transcription, and chromatin modification, suggesting that each metabolite may have a unique biological function [51]. Collectively, these results suggest that 5-mC oxidized derivatives function as stable modifications to modulate biological activity independent of their role as demethylation intermediates.

## 4. TDG in Cancer

TDG’s involvement in p53 signaling was one of the earliest indications of TDG as a potential tumor suppressor. TDG was found to potentiate p53 signaling, which in turn regulates its own expression [52]. TDG was also found to be essential for the expression of several tumor suppressors genes in vitro, such as *p15^ink4b^*, *Hic1*, *Rarβ*, and *Nr0b2* [29,53,54]. Several conditional *Tdg* knockout studies performed in mice have since provided support for TDG as a tumor suppressor in vivo [53,55]. One study demonstrated that intestinal-specific loss of *Tdg* in *Apc^Min^* mice, a well-characterized model of tumor disposition, resulted in a two-fold increase in small intestinal adenomas [55]. This phenotype was observed predominantly in female mice, suggesting that sexual dimorphism may contribute to cancer incidence in response to *Tdg* loss. Utilizing a novel conditional *Tdg* knockout mouse model containing the tamoxifen-inducible *Cre-ERT2* to excise *Tdg* in all tissues in a temporal fashion [53], our lab demonstrated that the conditional deletion of *Tdg* in adult mice (*Tdg*_cKO_) resulted in the development of late-onset hepatocellular carcinoma (HCC) and hepatoblastoma (HB). Interestingly, a sex bias in HCC incidence in the *Tdg*_cKO_ mice was observed, with male mice displaying an approximately 2-fold increase in HCC incidence compared to females. Moreover, male *Tdg*_cKO_ mice displayed increased body weight and glucose intolerance, which are common symptoms associated with obesity and type 2 diabetes, which are major risk factors for HCC [56]. Etiologically, loss of BA homeostasis is a major driver for HCC development in mice and humans [57]. Accordingly, male *Tdg*_cKO_ mice display increased hepatic and serum bile acids (BAs) with age. Immunohistochemistry of *Tdg*_cKO_ livers found that 5-caC staining was more intense in a subpopulation of cells in the liver. This provides evidence that the deletion of *Tdg* may block active DNA demethylation, leading to an accumulation of 5-caC in the liver in addition to loss of co-activator recruitment and associated transcriptional consequences. The observed phenotypes in *Tdg*_cKO_ mice have been recapitulated using a liver-specific *Tdg* knockout mouse model (unpublished observations). Through high-throughput transcriptomic analysis of male *Tdg*_cKO_ livers followed by gene-set enrichment analysis, metabolism was identified as the most dysregulated pathway in *Tdg*_cKO_ mice. Considering that the Farnesoid X Receptor (FXR) is the master regulator of diverse metabolic processes, including hepatic BA and glucose metabolism, it is likely that TDG’s coactivating role in FXR signaling plays a considerable role in the maintenance of hepatic homeostasis. Similar to *Tdg*_cKO_ mice, *Fxr* knockout mice also develop a late-onset HCC and display symptoms associated with obesity and type 2 diabetes, including glucose intolerance and the accumulation of primary bile acids with age [58,59]. Importantly, intraperitoneal injection of mice with an FXR agonist GW4064 caused rapid recruitment of an FXR complex consisting of FXR, TDG, the lysine acetyltransferase CBP, and TET2 to a subset of FXR target genes (Figure 2). Collectively, these findings demonstrate that a loss of *Tdg* leads to a dysregulation in the FXR-SHP axis in the liver and that *Tdg*_cKO_ mice exhibit an increased prevalence of HCC in a background of elevated serum and intrahepatic BAs. The onset of liver cancer in *Tdg*_cKO_ mice was surprising considering that expression of *Tdg* is ubiquitous and is in stark contrast to *Tet* knockouts, which result in predominantly hematopoietic abnormalities and malignancies [60]. Unlike the hematopoietic system, hepatocytes, under normal physiological conditions, are mitotically dormant and mostly found in the quiescent state (G_0_). This may provide a more favorable environment for the accumulation of 5-fC/5-caC in *Tdg*_cKO_ livers during active DNA demethylation, which is a replication-independent process.

In contrast to its tumor suppressive properties, two studies have shown that TDG can promote tumourigenesis and may be a potential target for cancer therapy. The first study showed that *Tdg* is overexpressed in a subset of human colorectal cancer (CRC) patients [61]. TDG acts as a positive regulator of WNT signalling by functioning as an adaptor protein for the transcription factor TCF4 and recruiting CBP/p300. Moreover, stable transfection of TDG shRNA into several CRC cell lines inhibited cell growth. Importantly, stable knockdown of *Tdg* reduces the ability of CRC cells to form tumors in xenograft assays suggesting that TDG is required for CRC cell proliferation in vivo. More recently, utilizing melanoma cell line models it was shown that inactivation of TDG causes cell cycle arrest and senescence along with increased DNA methylation at a subset of CpG sites [62]. Furthermore, *Tdg* knockdown was shown to supress tumor formation of melanoma cell lines in xenograft models suggesting that TDG activity is critical for tumor induction and/or progression. Using a high throughput screening assay dependent on TDG catalytic activity, the authors identified first generation TDG inhibitors that decreased viability and clonogenic capacity of melanoma lines.

To date no homozygous mutations have been identified in cancer patients. A heterozygous missense mutation in *Tdg* that is associated with reduced TDG protein levels has been identified in rectal cancer [63]. In humans, several single nucleotide polymorphisms (SNPs) in the *Tdg* gene have been associated with an increased risk of cancer development. For example, the SNP rs4135054 is associated with esophageal squamous cell carcinoma (ESCC) [64]. Additionally, a non-synonymous coding SNP rs2888805 (V367M mutation) and an intronic SNP rs4135150 are associated with an increased risk of developing non-melanoma skin cancer (NMSC) and other cancers [65]. A more recent study determined that two other SNPs (rs4135113 and rs1866074) are associated with an increased risk of colorectal cancer [66]. The AA genotype of the SNP rs4135113 increased the risk of colon cancer development by more than 3.6-fold, whereas the minor allele A increased the risk by 1.6-fold. Collectively, these findings suggest that TDG possesses both tumor suppressive properties as well as oncogenic properties depending on the type of cancer involved (Table 1).

## 5. Deregulated Active DNA Demethylation and Cancer

Not surprisingly, the genome-wide distribution of 5-mC derivatives is dramatically altered in most cancers compared to normal tissues, which raises the possibility that dysregulation of the DNA demethylation machinery may lead to a reprogramming of the epigenomic landscape in cancer. 5-hmC levels are dramatically reduced in several cancer types including breast, liver, lung, gastric, prostrate, pancreatic, renal, as well as glioblastoma and melanoma [67,68,69,70,71,72,73]. This reduction in 5-hmC largely occurs within gene bodies and regulatory regions such as enhancers and transcription factor binding sites, which may account for promoter hypermethylation observed in most cancers [74]. Low 5-hmC levels serve as a predictive marker for poor prognosis and survival in several cancers and is also correlated with decreased expression of TETs [75]. Furthermore, TET mutations are common in several types of cancers, and the DNA methylation patterns found in TET-deficient cells are similar to those of cancer genomes showing promoter hypermethylation in combination with widespread hypomethylation within heterochromatin [76]. Measurement of 5-fC in ES cells using a chemical pulldown approach has shown that upon *Tdg* knockdown, the gene promoters that showed the largest increase in 5-fC tended to gain methylation during differentiation of ES cells suggesting that normal 5-fC excision may be critical for the establishment of correct methylation patterns [77]. We have shown that the Hypermethylated in cancer 1 (*Hic1*) gene undergoes active DNA demethylation in response to retinoic acid and that the loss of active DNA demethylation precedes hypermethylation and silencing of *Hic1* in the same tissue [29].

## 6. Future Perspectives

The recent observation that TDG can alter chromatin structure through direct interaction with DNA, as well as functioning in long range chromatin fiber interactions, adds an additional layer of complexity to the multi-faceted nature of this protein. However, the chromatin remodeling activity has been identified largely using in vitro assays. Additional studies are necessary using genomic approaches based on chromosome capture technology to examine the dependency of the spatial organization of the genome on TDG. The N- and C-terminal regions of TDG are essential for interactions with other proteins and contain important sites for post-translational modifications such as acetylation, phosphorylation, and SUMO conjugation [78]. While it has been suggested that SUMO conjugation causes the release of TDG from abasic sites [79,80], the exact role for these covalent modifications in active DNA demethylation remains unclear and should be explored. Moreover, further studies should investigate potential mechanisms contributing to the sexual dimorphism in cancer development observed in both conditional *Tdg* knockout mouse models. TDG’s coactivating role in various sex hormone-related pathways (i.e., androgen signaling) may likely have a contributing effect to these sex differences. The discovery of first-generation inhibitors of TDG suggests that TDG is a druggable target in cancer. These inhibitors should be tested on other in vivo models to determine their efficacy for treating other types of cancers. Alternatively, developing molecular tools to introduce site-specific DNA demethylation at hypermethylated, cancer-associated loci may prove to be beneficial as a potential cancer therapy [81]. Further research should be conducted regarding TDG’s role in FXR signaling and hepatic homeostasis, as well as the role of 5-fC and 5-caC in cancer regarding their use as potential biomarkers for various cancers.

## 7. Conclusions 

TDG has key functions in DNA repair, DNA demethylation, and as a transcriptional co-activator. These functions play overlapping roles in gene regulation through associations with various interacting partners. It is becoming evident that active DNA demethylation is a critical aspect of gene regulation that has important ramifications in cancer research. The recent conditional *Tdg* knockout studies have demonstrated TDG’s role as a tumor suppressor in vivo and as a tumor promoter in some contexts, highlighting the importance of TDG’s role for maintaining normal cellular homeostasis. 

## Figures and Tables

**Figure 1 cancers-14-00765-f001:**
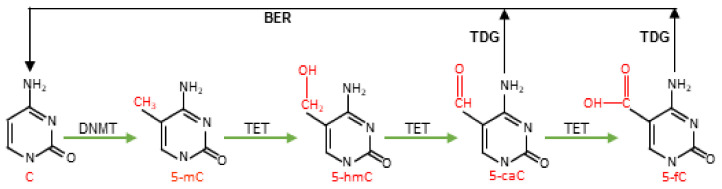
Active DNA demethylation pathway. DNA methyltransferases (DNMT) methylate unmodified C to generate 5-methylcytosine (5-mC), which can be successively oxidized by ten eleven translocation (TET) enzymes to generate 5-hydroxymethylcytosine (5-hmC), 5-formylcytosine (5-fC), and 5-carboxylcytosine (5-caC). Highly oxidized cytosine derivatives, 5-fC and 5-caC, are excised by Thymine DNA Glycosylase (TDG) and repaired through base excision repair (BER) to regenerate unmodified C.

**Figure 2 cancers-14-00765-f002:**
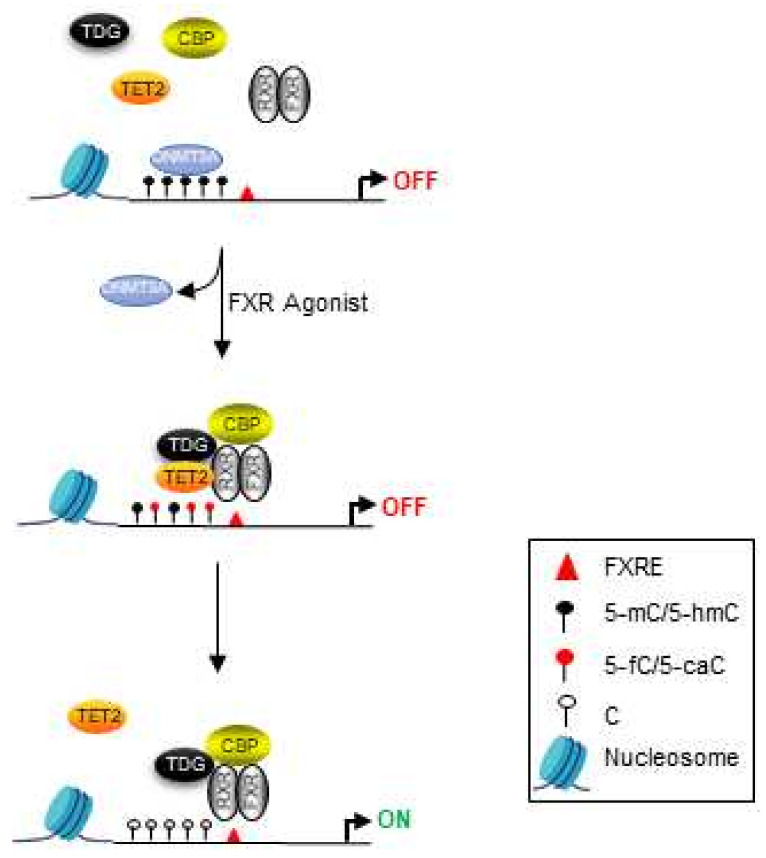
Transcriptional regulation of FXR-target genes by TET/TDG-mediated active DNA demethylation. In the absence of ligand, DNMT3A is bound to methylated DNA in a transcriptionally inactive state. In the presence of FXR agonist, the FXR/RXR heterodimer recruits TDG, CBP, and TET2 to form a ternary complex at target genes. DNMT3A is displaced and 5-mC undergoes oxidation to 5-fC/5-caC in a TET2/TDG dependent manner. TDG excises 5-fC and 5-caC leading to restoration of the unmethylated cytosine and transcriptional activation.

**Table 1 cancers-14-00765-t001:** Differing roles of TDG in various cancers.

Cancer	Species	Role of TDG	Phenotype/Effect	References
HCC/HB	Mouse	Tumor suppressor	Loss of *Tdg* results in increased HCC/HB incidence predominantly in male mice	[53]
Intestinal adenoma	Mouse	Tumor suppressor	Loss of *Tdg* results in a two-fold increase in small-intestinal adenomas predominantly in female mice	[55]
CRC	Human	Oncogene	*Tdg* expression is upregulated in human CRC	[61]
	Human	-	Two SNPs (rs4135113/rs1866074) are associated with increased risk of CRC	[66]
	Mouse	Oncogene	*Tdg* knockdown inhibits xenografted CRC growth in nude mice	[61]
Melanoma	Human	Oncogene	*Tdg* knockdown/inhibition reduces viability of melanoma cell lines	[62]
Rectal cancer	Human	Tumor suppressor	Decreased TDG expression due to D284Y mutation is associated with increased risk of rectal cancer	[63]
ESCC	Human	-	SNP (rs4135054) is associated with increased risk of ESCC	[64]
NMSC	Human	-	Two SNPs (rs288805/rs4135150) are associated with increased risk of NMSC	[65]

HCC—hepatocellular carcinoma; HB—hepatoblastoma; CRC—colorectal cancer; ESCC—esophageal squamous-cell carcinoma; NMSC—nonmelanoma skin cancer.

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
