# Peer review of "The Role of Thymine DNA Glycosylase in Transcription, Active DNA Demethylation, and Cancer"

_cancers, 2022, doi:10.3390/cancers14030765_

Round 1
Reviewer 1 Report
The manuscript by Oladapo Onabote et al. “The role of Thymine DNA Glycosylase in transcription, active demethylation, and cancer” provides a short review of recent advances in understanding of the role of mismatch-specific DNA glycosylase in various biologically important processes. Authors briefly review mechanisms of active DNA demethylation and the role of TDG in transcription regulation. Then authors described in more details the role of TDG in both tumour suppression and cancer progression. In conclusion, this is an up-to-date review. Apart from few minor comments, this review is important for researchers working in fields of DNA repair, methylation and cancer. I have some minor revisions to suggest:
Minor comments.
1. Throughout the text authors referred to the genomic 5-methylcytosine cleansing as “active demethylation”, which might be misleading for a non-specialist reader. Actually, histone methylation and demethylation are active enzymatic processes playing important roles in the epigenetic regulation of cellular processes. Therefore, “active DNA demethylation” should be used instead of “active demethylation” in order to separate histones and other cellular macromolecules from DNA.
2. In Introduction section authors wrote: “… in the context of CpG dinucleotides that are often clustered together in stretches of DNA called CpG islands. CpG islands can be found within promoter, enhancer, intergenic and intronic regions and are often associated with transcriptional regulation of associated genes. …”
This description lacks some important details necessary for understanding the biological function of TDG. For example, it is important to note that vertebrate CpG islands are short interspersed CG-rich genomic regions that were identified at the very beginning as unmethylated fraction of genomic DNA (Bird, et al. 1985). In mammalian genomes, CpG islands are typically 500-3,000 base pairs in length, and have been found in or near half of the promoters of mammalian genes (Akan and Deloukas 2008). Gene promoters of all housekeeping genes are associated with CpG islands, they are enriched in permissive histone modifications, poor in DNA cytosine methylation and contain multiple sites for transcription factors (Deaton and Bird 2011).
3. In Introduction section authors wrote : “… In this process, the ten eleven translocation (TET) family of proteins recognize and successively oxidize 5-mC to …”
Authors should note that in mammals, three functional TET paralogs have been identified: TET1, TET2, and TET3 (Iyer LM et al., Cell Cycle. 2009;8:1698 ; Pastor WA et al., Nat. Rev. Mol. Cell Biol. 2013;14:341; Williams K et al., EMBO Rep. 2012;13:28). In mammalians before their diversification the TET genes underwent two successive duplications to give rise three distinct proteins (Iyer LM et al., Cell Cycle. 2009;8:1698 ; Pastor WA et al., Nat. Rev. Mol. Cell Biol. 2013;14:341).
4. When describing the DNA substrate specificity of TDG it would nice to mention the repair of 7,8-dihydro-8-oxoadenine (8oxoA) described in (Jensen A et al., J Biol Chem 278, 19541, 2003; Talhaoui I et al., Nucleic Acids Res 41, 912, 2013).
5. Page 4, section N°3 TDG in active demethylation. Following sentence is written in non-standard English: “… demonstrating that GADD45a enhances DNA demethylate on by TDG [39].”
6. Page 8, section N°6 Conclusions. Authors wrote: “These inhibitors should be tested on other in vivo models to test their efficacy for treating other types of cancers.”
Authors should avoids tautology, repeating word “test”.
Author Response
Reviewer 1 Minor Comments:
- Throughout the text authors referred to the genomic 5-methylcytosine cleansing as “active demethylation”, which might be misleading for a non-specialist reader. Actually, histone methylation and demethylation are active enzymatic processes playing important roles in the epigenetic regulation of cellular processes. Therefore, “active DNA demethylation” should be used instead of “active demethylation” in order to separate histones and other cellular macromolecules from DNA.
- Every mention of “active demethylation” was changed to “active DNA demethylation”
- In Introduction section authors wrote: “… in the context of CpG dinucleotides that are often clustered together in stretches of DNA called CpG islands. CpG islands can be found within promoter, enhancer, intergenic and intronic regions and are often associated with transcriptional regulation of associated genes. …”
This description lacks some important details necessary for understanding the biological function of TDG. For example, it is important to note that vertebrate CpG islands are short interspersed CG-rich genomic regions that were identified at the very beginning as unmethylated fraction of genomic DNA (Bird, et al. 1985). In mammalian genomes, CpG islands are typically 500-3,000 base pairs in length, and have been found in or near half of the promoters of mammalian genes (Akan and Deloukas 2008). Gene promoters of all housekeeping genes are associated with CpG islands, they are enriched in permissive histone modifications, poor in DNA cytosine methylation and contain multiple sites for transcription factors (Deaton and Bird 2011).
- The following edits to the Introduction section were made to the manuscript: “5-mC occurs predominantly in the context of CpG islands, which are short, interspersed, CG-rich regions of DNA that were originally identified as unmethylated fraction of mouse genomic DNA [2]. CpG islands are typically 500-3,000 base pairs in length and are found within or proximal to nearly half of the promoters of mammalian genes [3]. For example, all housekeeping genes are associated with promoter CpG islands that are enriched in permissive histone marks, contain multiple transcription factor-binding sites, and are typically unmethylated [4]. CpG islands can also be found within enhancer, intergenic and intronic regions and are often associated with transcriptional regulation of associated genes.”
- In Introduction section authors wrote : “… In this process, the ten eleven translocation (TET) family of proteins recognize and successively oxidize 5-mC to …”
Authors should note that in mammals, three functional TET paralogs have been identified: TET1, TET2, and TET3 (Iyer LM et al., Cell Cycle. 2009;8:1698 ; Pastor WA et al., Nat. Rev. Mol. Cell Biol. 2013;14:341; Williams K et al., EMBO Rep. 2012;13:28). In mammalians, before their diversification the TET genes underwent two successive duplications to give rise three distinct proteins (Iyer LM et al., Cell Cycle. 2009;8:1698; Pastor WA et al., Nat. Rev. Mol. Cell Biol. 2013;14:341).
- The following edits to the Introduction were made to the manuscript: “Alternatively, 5-mC can be removed in a process called active DNA demethylation that is independent of the cell cycle. Rather, this process is dependent on the oxidative function of the ten eleven translocation (TET) family of proteins, of which there are three functional paralogs (TET1, TET2, and TET3) that exist in mammals due to a gene triplication event that occurred in jawed vertebrates.”
- When describing the DNA substrate specificity of TDG it would nice to mention the repair of 7,8-dihydro-8-oxoadenine (8oxoA) described in (Jensen A et al., J Biol Chem 278, 19541, 2003; Talhaoui I et al., Nucleic Acids Res 41, 912, 2013).
- 7,8-dihydro-8-oxoadenine (8oxoA) was included to the list of substrates
- Page 4, section N°3 TDG in active demethylation. Following sentence is written in non-standard English: “… demonstrating that GADD45a enhances DNA demethylate on by TDG [39].”
- “GADD45a enhances DNA demethylation by TDG”
- Page 8, section N°6 Conclusions. Authors wrote: “These inhibitors should be tested on other in vivo models to test their efficacy for treating other types of cancers.” Authors should avoids tautology, repeating word “test”.
- The word “test” has been changed to “determine”
Reviewer 2 Report
This review is well written and comprehensive. My only comment is to advise the authors to include a diagram illustrating the functions of TDG.
Author Response
This review is well written and comprehensive. My only comment is to advise the authors to include a diagram illustrating the functions of TDG.
- Two figures illustrating TDG’s roles in active DNA demethylation and transcriptional regulation are included.
Reviewer 3 Report
In this manuscript, the authors performed a systematic review on the role of thymine DNA glycosylase (TDG) in transcription, active demethylation, and cancer. The manuscript is well-written. The introduction section described in detail the background information of methylation of cytosine at the 5th carbon (5-mC) and mechanisms of involved in removal of the 5-mC mark. Then, the authors discussed the role of the TDG in transcriptional regulation and recent studies using animal models which establish a connection between TDG and tumorigenesis in vivo. Finally, the authors provided an overview of the mechanistic work that has been documented on the role of TDG in gene regulation as a framework for understanding its physiological relevance in cancer. I only have several minor suggestions.
- A general objective of a review paper is comprehensive but easy to follow. The authors described too many details of mechanisms and published studies, which are good but a little bit hard to follow. I strongly suggest authors use figures to illustrate where applicable, especially when complex mechanisms are presented.
- The perspective part is relatively short and should be strengthened. The authors need to propose several further research directions based their conclusions.
- Too many abbreviations are used which also makes the manuscript hard to follow. A section or a supplementary table describing all abbreviations used throughout the manuscript should be provided.
Author Response
- A general objective of a review paper is comprehensive but easy to follow. The authors described too many details of mechanisms and published studies, which are good but a little bit hard to follow. I strongly suggest authors use figures to illustrate where applicable, especially when complex mechanisms are presented.
- Figures illustrating TDG’s roles in active DNA demethylation and transcriptional regulation are included.
- The perspective part is relatively short and should be strengthened. The authors need to propose several further research directions based their conclusions.
- We have expanded the Perspective section by proposing additional research which should be conducted regarding the role of TDG in gene regulation and the role of TDG in sexual dimorphism and cancer development.
- Too many abbreviations are used which also makes the manuscript hard to follow. A section or a supplementary table describing all abbreviations used throughout the manuscript should be provided.
- A Supplementary table containing the list of abbreviations has been included.